# Proteomics-Based Regression Model for Assessing the Development of Chronic Lymphocytic Leukemia

**DOI:** 10.3390/proteomes9010003

**Published:** 2021-01-23

**Authors:** Varvara I. Bakhtina, Dmitry V. Veprintsev, Tatiana N. Zamay, Irina V. Demko, Gleb G. Mironov, Maxim V. Berezovski, Marina M. Petrova, Anna S. Kichkailo, Yury E. Glazyrin

**Affiliations:** 1Department of Hematology, Krasnoyarsk Regional Clinical Hospital, 660022 Krasnoyarsk, Russia; doctor.gem@mail.ru (V.I.B.); demko64@mail.ru (I.V.D.); 2Laboratory for Digital Controlled Drugs and Theranostics, Federal Research Center “Krasnoyarsk Science Center of the Siberian Branch of the Russian Academy of Science”, 660036 Krasnoyarsk, Russia; d_veprintsev@mail.ru (D.V.V.); annazamay@yandex.ru (A.S.K.); 3Laboratory for Biomolecular and Medical Technologies, Krasnoyarsk State Medical University Named after Prof. V.F. Voyno-Yasenetsky, 660022 Krasnoyarsk, Russia; tzamay@yandex.ru; 4Faculty of Medicine, Krasnoyarsk State Medical University Named after Prof. V.F. Voyno-Yasenetsky, 660022 Krasnoyarsk, Russia; stk99@yandex.ru; 5Department of Chemistry and Biomolecular Sciences, University of Ottawa, Ottawa, ON K1N6N5, Canada; ggmironov@gmail.com (G.G.M.); Maxim.Berezovski@uottawa.ca (M.V.B.)

**Keywords:** chronic lymphocytic leukemia, CLL, linear regression, proteomics, mass spectrometry, label-free quantification

## Abstract

The clinical course of chronic lymphocytic leukemia (CLL) is very ambiguous, showing either an indolent nature of the disease or having latent dangerous progression, which, if diagnosed, will require an urgent therapy. The prognosis of the course of the disease and the estimation of the time of therapy initiation are crucial for the selection of a successful treatment strategy. A reliable estimating index is needed to assign newly diagnosed CLL patients to the prognostic groups. In this work, we evaluated the comparative expressions of proteins in CLL blood cells using a label-free quantification by mass spectrometry and calculated the integrated proteomic indexes for a group of patients who received therapy after the blood sampling over different periods of time. Using a two-factor linear regression analysis based on these data, we propose a new pipeline for evaluating model development for estimation of the moment of therapy initiation for newly diagnosed CLL patients.

## 1. Introduction

Chronic lymphocytic leukemia (CLL) is a malignant blood disease that is characterized by a progressive accumulation of functionally incompetent CD5- and CD19-positive mature B-type lymphocytes [1]. CLL is the most common form of leukemia in adults in Western countries [2]. The understanding of the pathological mechanisms involved in the CLL formation helps to divide the disease into subgroups, which are considerable for prognostication and treatment. Despite detailed studies of clinical features and chromosomal abnormalities in CLL, the molecular details underlying disease development are still not entirely clear [3].

The clinical course of patients with CLL is extremely heterogeneous. While in some cases the disease has an indolent behavior and patients eventually die because of causes not related to the disease, in others it shows an aggressive clinical course and patients die shortly after diagnosis, due to the disease or treatment-related complications. The lifespan of patients with a diagnosed CLL ranges from less than 1–2 years to more than 15 years [4]. The widely used Binet and Rai staging systems [5,6] have proved to be invaluable in predicting the clinical outcome between the various staging groups. However, they are unable to identify prognostic groups with good and bad outcome within the stage [7]. As most patients come to clinical attention in the early stage of the disease and do not require immediate treatment, the future therapy necessity and estimation of the time of therapy initiation are relevant.

Diagnostic biomarkers serve as indicators of the presence and activity of pathogenic processes in organism in general [8]. Unlike diagnostic biomarkers, prognostic biomarkers allow one to observe the process development over time and denote natural history of disease, which is independent from the treatment [9]. Prognostic markers can predict patient’s lifespan in aggressive state of disease, or they might help to evaluate a point of future initiation of treatment in the case of a latent state of CLL [10].

The following molecular-based CLL prognostic markers are currently used in clinical practice: (1) immunoglobulin heavy-chain variable region (IgHV) mutational status, (2) interphase fluorescence in-situ hybridization (iFISH) abnormalities, (3) cluster of differentiation (CD) 38, and (4) zeta-associated protein (ZAP)-70 [10]. However, none of these markers provides the direct estimation of time before therapy initiation; moreover, they still require standardization and validation [4]. The mechanisms by which these markers influence a disease progression are under investigation as well [11].

Mass spectrometry-driven proteomics generates a large amount of biologically relevant data. Recent advances in data mining technologies can make this approach more applicable in medicine. For instance, the linear regression analysis was presented as a powerful estimating instrument for the prediction of heart diseases based on multiple clinical indicators [12]. However, so far, no published example of the application of linear regression based on proteomic data for medical estimating purposes has been found in the literature.

To develop a new evaluating model using a two-factor linear regression to estimate the time of the therapy initiation, we used the results of the proteomic profiling of blood lymphocytes from patients with diagnosed CLL. Blood cells sampling was performed prior to treatment, but all the patients were treated later at different times, which allowed us to use this observed time parameter to improve the quality of the evaluating model. The quantitative data on the cellular proteins were obtained by high-resolution mass spectrometry and calculated by label-free quantification (LFQ) approach. The best regression models based both on direct LFQ data for certain proteins as well as integrated full-proteomic indexes expressed in principal components are presented here.

## 2. Materials and Methods

### 2.1. Sampling

This study was approved by the Local Ethics Committee of the Krasnoyarsk State Medical University named after Professor V.F. Voyno-Yasenetsky (ethical code 37/2012 of 31 January 2016) and was conducted in accordance with the principles of the Declaration of Helsinki. All patients gave their informed consent for inclusion before they participated in the study.

Lymphocytes isolation from blood was performed by Lympholyte-H (Cedarlane, Burlington, ON, Canada) according the manufacturer’s protocol. Before further processing, the cell samples were stored in a Bambanker cryoconservation medium (Lymphotec, Tokyo, Japan) at −80 °C. After that, the suspension of cells was washed and lysed in a 0.1% solution of sodium deoxycholate for 30 min. The protein concentrations in supernatant obtained after centrifugation were measured by UV-1280 spectrophotometer (Shimadzu, Kyoto, Japan). The equivalent amount of 4 μg of protein per sample was taken for mass spectrometry. The protein samples were reduced, alkylated, and digested by trypsin using the universal reagent set from Thermo Scientific (Waltham, MA, USA). The samples were cleaned by 10-μL C18 pipette tips and dried before analysis. All samples were prepared and analyzed in triplicates.

### 2.2. Chromatography and Mass Spectrometry

The samples were resuspended in phase “A” (0.1% of formic acid) before analysis. The gradient from 0% to 40% of phase “B” (0.1% of formic acid in 80% acetonitrile) was run for 120 min using a Dionex UltiMate 3000 RSLC nano liquid chromatographer (Thermo Scientific, Waltham, MA, USA). The Acclaim RLSC PepMap C18 column (75 μm inner diameter, 2 μm particles, 15 cm length) was used. The nano-flow was set to 200 nL/min. The Orbitrap Fusion mass spectrometer (Thermo Scientific, Waltham, MA, USA) was set to data dependent mode with alternating scans of parent and fragment ions. Resolution of 60,000 was set for first scans made by an Orbitrap mass detector. Subsequent fragment ions were analyzed after collision-induced dissociation by ion trap detector at a normal rate.

### 2.3. Protein Identification and Label-Free Quantification

The total set of mass spectrometry files was processed by MaxQuant 1.5 software (Max Planck Institute for Biochemistry, Martinsried, Germany) [13] with enabled label-free quantification (LFQ). Only unique peptides were chosen for protein quantification. Cysteine carbamidomethylation, methionine oxidation, and N-term acetylation were chosen as possible modifications. The reviewed Swiss-Prot protein database was used for protein search with a false discovery rate of 0.01. The MaxQuant outputs relative values expressed in LFQ intensities for the identified proteins, which are considered as quantitative indicators of protein distribution over the sample groups.

### 2.4. Data Processing

The proteins identified by MaxQuant were excluded only in one sample. The quantitative data for each protein were normalized by dividing the LFQ values by the maximum of the LFQ value of the protein in all samples. A two-parameter linear model was developed for each protein:T = α · T_0_ + β ∙ LFQ(Protein) + γ,(1)
where T is a period of time from sampling to therapy initiation and T_0_ is a period of time from diagnostics to sampling (both in months). To estimate the quality of the models we used the following well-known metrics: the coefficient of determination *R*^2^ as well as the *F*-values and *p*-values corresponding to the model coefficients.

In addition, a linear regression model T on T_0_ and P_1_ was built,
T = α · T_0_ + β ∙ P_1_ + γ,(2)
where P_1_ is a projection of the vector of the LFQ values to the first principal component. In order to estimate the adequacy of this model, in addition to standard metrics, we used leave-three-out cross-validation [14]—the procedure of multiple calculations of the quality metrics using different splits of the data set to train/test sets.

All steps of data analysis were performed in Anaconda Python 3.

## 3. Results

The experimental group (Table 1) consisted of five B-cell CLL patients (average age was 60.6 years). None of these patients received any treatment at the time of sampling (July 2014). After sampling, they received therapy at different time intervals (column T in Table 1). Two patients received Leukeran therapy and three patients received fludarabine, cyclophosphamide, and rituximab therapy. Most of them had negative test on CD38 and ZAP-70, which are known as markers of adverse course of the disease.

The difference in the time after diagnosis and before the start of the treatment of the patients included in the study allowed us to construct evaluating models using individual data of proteomic cell profiling. The model quality was assessed by comparing the evaluated moments of therapy initiation to the observed values for the entire group of patients.

The comparative LFQ numbers were obtained for 2128 proteins from the total set of mass spectra using the MaxQuant 1.5 program (Appendix A). After filtration, the final set consisted of 1183 proteins. Then the LFQ values were normalized.

After applying linear regression in the following form:T = α · T_0_ + β ∙ LFQ(Protein) + γ,(3)
where T is the period of time from sampling to therapy initiation and T_0_ is the period of time from diagnostics to sampling (both in months), models were evaluated based on the LFQs of single proteins according to the highest values of the coefficients of determination *R*^2^ that were obtained. Proteins selected with the best fit for the model (1) are presented in Table 1. These proteins are common for all five of the patients that were tested (Appendix A).

The *F*-values of the models were less than 10^−10^; the *p*-values corresponding to the model coefficients were less than 10^−8^.

To obtain a general indicator for our data set that was not based on individual candidate biomarkers, while taking into account the contribution of all proteins, we used the method of lowering the dimensions by principal component analysis (PCA). The projection P_1_ onto the first principal component for the LFQ vector of each sample was found. Then a linear regression model was built:T = α · T_0_ + β ∙ P_1_ + γ,(4)

Taking T and T_0_ in months, the values of the coefficients α, β, and γ were −0.83, −1.78, and 46.27, respectively. The value of the coefficient of determination *R*^2^ was 0.99. The *F*-value of the model was 3.3 × 10^−15^, and the *p*-value corresponding to the model coefficients was less than 10^−12^. In order to estimate the adequacy of the linear model of the form (2) in general, we performed leave-three-out cross-validation. For this, we sequentially removed the certain patient’s data from the dataset, fitted the model, and estimated the model quality based on the removed data (test dataset). It was found that the average value of *R*^2^ on the test dataset was 0.95 and the average value of mean squared error was 1.4. Thus, the model (2) (Figure 1) can be recommended for further testing on larger sets of samples.

## 4. Discussion

We obtained the best evaluating regression models for estimating the rate of CLL progression with the *F*-values less than 10^−10^, and the *p*-values less than 10^−8^, based on the expression levels of individual proteins (Table 2). All of the three proteins perform key cellular functions. Mitochondrial 60 kDa heat shock protein plays a significant role in carcinogenesis, promotes the proliferation of cancer cells, and serves as an antiapoptotic mediator for various types of cancer [15,16]. The content of 40S ribosomal protein SA increases with various malignant tumors, in particular melanoma, colon cancer, breast and esophageal cancer, and glioma [17,18,19,20,21]. Polypyrimidine tract-binding protein 1 promotes the development of metastases with colorectal cancer and clear-cell renal cell carcinoma, glioblastoma, and epithelial ovarian tumors [22,23,24,25,26].

Although the individual proteins presented in the best evaluating models are responsible for the active life cycle and proliferation of cancer cells and may be related to the whole oncogenesis, they were not mentioned as specific markers of CLL. The function of any distinct predictive biomarker is not entirely relevant in an unbiased approach. A long list of various proteins may be involved in the origin and course of CLL. Therefore, we should consider models that take into account the contributions of larger sets of proteins that affect pathology. At best, it should be the whole proteome found in a cell sample.

The most proper strategy for identifying individual markers of malignancy would be proteomic comparison of CLL lymphocytes with healthy cells. In our case, we used new integrative parameters as a diagnostic feature. The new principal components take into account the contribution of all proteins registered in mixtures of normal and malignant lymphocyte lysates. Wherein, we assume that the main contribution to the principal components is made by the proteins that are most varied in different samples. Typically, these are proteins specific to the malignant cells. Normal proteins, which originate from the healthy lymphocytes, and do not differ in the level of expression in different patients, should not contribute to the principal components and are not taken into account in the model.

The regression model (2) (Figure 1), showing the *F*-value of 3.3 × 10^−15^ and the *p*-value less than 10^−12^, turned out to be the preferred one, since it considered the contribution of all the proteins together. Moreover, this model did not depend on a single biomarker. Therefore, it may be more reliable and universal, and may be applicable for the clinical prognosis of the disease and evaluation of the treatment effectiveness.

The use of the presented approach considers the concept of a clinical test system, where the separation of malignant cells from normal lymphocytes using antibodies seems to be a laborious task at the routine level. The idea of the new test system was to take lymphocytes “as is” to simplify the clinical procedure. In future, the test for evaluating the time of the start of therapy can be performed in a laboratory equipped with a mass spectrometer. Data on the proteomic profile of blood cells from a newly diagnosed patient should be analyzed together with the data of known patients. When adding the new data into an existing regression model based on data from known patients, it will be possible to assess the moment of the therapy initiation for a new patient. With an increase in the samples used, and with further verification in the future, the reliability of the method will increase. However, the results obtained in our work are based on a small number of patients in each group. Therefore, in this form, they can only be a successful demonstration of the proof-of-concept. Further verification on larger sets of samples is definitely required.

## 5. Conclusions

Here, we propose an approach based on linear regression, which can be used for clinical purposes for evaluating disease progression. For this, a primary set of proteomic and time-related data for the regression model development should be collected. The newly diagnosed patient’s proteome should be analyzed on the same instrumental platform, and the new data should be processed along with the primary dataset in order to obtain relative quantitative proteomic values. The obtained regression model could be used for the assessment of disease progression. We introduce a concept of the evaluating system for CLL, based on a small-scale dataset. A more accurate prognostic model can be developed along with the accumulation of more extended proteomic and time-related data based on blood cancer clinical records.

## Figures and Tables

**Figure 1 proteomes-09-00003-f001:**
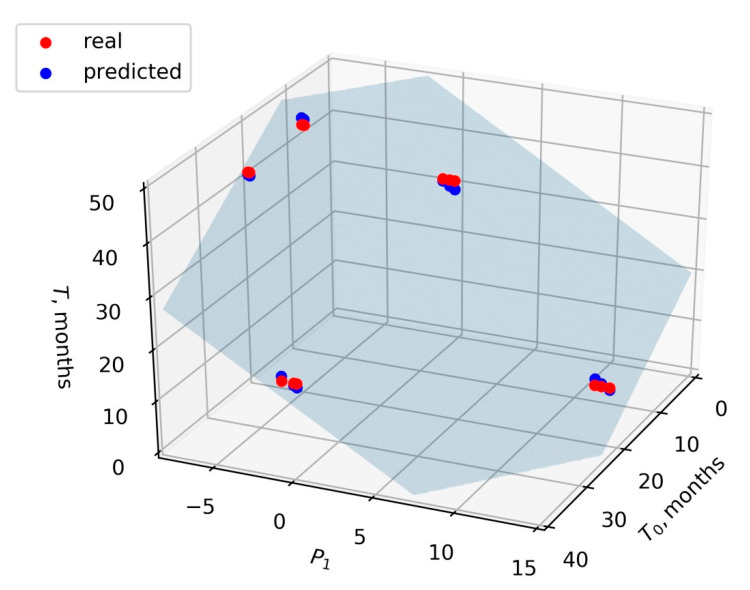
A plane according to the model (2), describing the dependence of T on T_0_ and on P_1_. Blue points are predicted values (lying in the model plane), red points are real samples. P_1_: projection of the vector of the label-free quantification (LFQ) values to the first principal component

**Table 1 proteomes-09-00003-t001:** Characteristics of patients. T_0_: period of time from diagnostics to sampling. T: period of time from sampling to therapy initiation. Traditional prognostic markers status: cluster of differentiation 38 (CD38) and zeta-associated protein-70 (ZAP-70).

Patient	Age, Years	Sex	T_0_, Months	T, Months	CD38	ZAP-70
XL-28	58	Male	9	37	Negative	Negative
XL-30	56	Male	13	46	Negative	Negative
XL-35	53	Female	22	42	Negative	Negative
XL-36	76	Male	38	16	Negative	Positive
XL-38	60	Male	17	8	Negative	No data

**Table 2 proteomes-09-00003-t002:** The best models according to the *R*^2^.

Protein ID	Protein Name	*R* ^2^	α	β	γ
P10809	60 kDa heat shock protein, mitochondrial	0.994	−1.39	−81.14	111.52
P08865	40S ribosomal protein SA	0.987	−0.80	−62.75	81.74
P26599	Polypyrimidine tract-binding protein 1	0.978	−0.98	−54.06	77.81

## Data Availability

The calculated data presented in this study are available in Appendix A. The raw mass spectrometry data presented in this study are available on request from the corresponding author.

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
