# Peer review of "Proteomics-Based Regression Model for Assessing the Development of Chronic Lymphocytic Leukemia"

_proteomes, 2021, doi:10.3390/proteomes9010003_

Round 1

Reviewer 1 Report

In "Proteomics-Based Prognostic Model for Predicting the Progression of CLL", the authors examined lymphocyte proteome in 5 CLL patients, in technical triplicate. They then analyzed the proteomics data for strong correlations between proteomics based peptide levels at T0 (months after patient diagnosed, but not yet treated) and the time T (months after diagnosis that therapy was initiated).

There is a strong clinical need for predictive molecular biomarkers in CLL, so that patients can be given the most appropriate treatment at the earliest appropriate time. An unbiased, proteomics-based approach holds high promise to contribute to this need. Having said that, the manuscript as presented needs revision prior to this reviewer being able to recommend it for publication. 

According to section 2.1 Sampling, the authors isolated lymphocytes from the blood of CLL patients, which contains both CLL cells and normal cells. While this is perfectly understandable, it would be valuable to compare the CLL samples to normal, non-diseased lymphocytes. This would allow for a significant enrichment of CLL-associated peptides through exclusion of normal peptide "contaminants". Normal controls for dataset training purposes are important for such a heterogeneous disease. 

In Table 2, it was unclear whether the proteins listed were common to all 5 patients, or whether they just represented the best R2 values from all of the data. This should be clearly explained.

In the discussion section, a case for why the three proteins from Table 2 are important for cancer is made. This felt superfluous, as in an unbiased approach, the function of predictive biomarkers isn't entirely relevant. 

No where in the paper was the word correlation used. The results were presented entirely as predictive and prognostic. This is the biggest concern for this reviewer. In order to be able to use the monikers "predictive" and "prognostic", the authors would need to predict the initiation of therapy in a proactive way, rather than a retrospective one, as they have done in this manuscript, with all of the variables provided (T0 and T). So, the authors need to alter the strength of the predictive claims in the paper, or show that in patients where T is unknown, that they are able to predict T accurately ahead of time.

While the manuscript in its current needs experimental and/or claim-cased revisions, this reviewer believes that potential of the the approach is exciting and holds much promise. 

Author Response

  1. According to section 2.1 Sampling, the authors isolated lymphocytes from the blood of CLL patients, which contains both CLL cells and normal cells. While this is perfectly understandable, it would be valuable to compare the CLL samples to normal, non-diseased lymphocytes. This would allow for a significant enrichment of CLL-associated peptides through exclusion of normal peptide "contaminants". Normal controls for dataset training purposes are important for such a heterogeneous disease. 

The authors agree with the reviewer. Proteomic comparison with healthy cells would be the most proper strategy for identifying individual markers of malignancy. In our case, we use new parameters as a diagnostic feature. The new principal components take into account the contribution of all proteins registered in mixtures of lymphocyte lysates. Wherein, we assume that the main contribution to the principal components is made by those proteins that are most varied in different samples. Typically, these are proteins specific to the malignant cells. Normal proteins, which originate from the healthy lymphocytes, and do not differ in the level of expression in different patients, should not contribute to the principal components and are not taken into account in the model.

In addition, our work considers the concept of a prototype clinical test system, where the separation of malignant cells from normal lymphocytes using antibodies seems to be a laborious task at the routine level. The idea of the new test system was to take lymphocytes “as is” to simplify the clinical procedure.

These remarks have been added to the Discussion.

  1. In Table 2, it was unclear whether the proteins listed were common to all 5 patients, or whether they just represented the best R2 values from all of the data. This should be clearly explained.

These proteins actually were selected from the best fit for the model (1) and represented the best R2 values from all of the data. However, they are also common to all 5 patients, as indicated in the Table S1. Notes on this issue have been added to the Results (line 148).

  1. In the discussion section, a case for why the three proteins from Table 2 are important for cancer is made. This felt superfluous, as in an unbiased approach, the function of predictive biomarkers isn't entirely relevant.

This part of the Discussion have been shortened and revised.

  1. No where in the paper was the word correlation used. The results were presented entirely as predictive and prognostic. This is the biggest concern for this reviewer. In order to be able to use the monikers "predictive" and "prognostic", the authors would need to predict the initiation of therapy in a proactive way, rather than a retrospective one, as they have done in this manuscript, with all of the variables provided (T0 and T). So, the authors need to alter the strength of the predictive claims in the paper, or show that in patients where T is unknown, that they are able to predict T accurately ahead of time.

The strength of the predictive claim in the paper has been reduced where possible.

Reviewer 2 Report

The authors describe a proteomics approach  to predict the time to  treatment  in CLL patients. They use cell lysates of isolated leucocytes from peripheral blood of 5 patients at different times before treatment. Of all patients there is only one time point. The approach is very interesting and the statistical data looks very strong. There is however no validation and given the “extremely heterogeneous” clinical course and the small experimental cohort readers will need some convincing this is applicable and how.

Some concerns:

  1. Can you show how the 3 proteins you used in your model behave in time with a figure?
  2. Please explain in the discussion what the approach to predict the time to treatment in individual patients would be, is it possible to perform this test in a standard lab?
  3. I would really like to see some kind of validation: within a patient in the time, is it suitable at diagnosis, in a small cohort of other patients?

Author Response

1. Can you show how the 3 proteins you used in your model behave in time with a figure?

Unfortunately, no supplemental sampling and proteome analysis were made after the start of treatment of the patients. Thus, it is impossible to trace the real level of expression of the presented proteins in the late stages of the disease in this experiment. This requires a new sampling process with repeated patient search, additional investment of time and resources. The meaning of this work was to predict the rate of the disease progression, based on the state of the proteome at an early stage only.

2. Please explain in the discussion what the approach to predict the time to treatment in individual patients would be, is it possible to perform this test in a standard lab?

The prospects of the method were briefly mentioned in the conclusions. Possibilities of applying the proposed approach in a standard laboratory have been added to the Discussion: “In prospect, the test for predicting the time of the start of therapy can be performed in a laboratory equipped with a mass spectrometer. Data on the proteomic profile of blood cells from a newly diagnosed patient should be analyzed together with the data of known patients. When adding the new data into an existing regression model based on data from known patients, it will be possible to assess the moment of the therapy initiation for a new patient. With an increase in the sample used, verified in the future, the reliability of the method will increase.”

3. I would really like to see some kind of validation: within a patient in the time, is it suitable at diagnosis, in a small cohort of other patients?

Validation is possible in the future with the addition of proteomic data from new patients. Unfortunately, this work is limited to the set of patients who were studied at the initial moment before the start of therapy. After that, a certain period of time passed and new data appeared about the need and initiation of the therapy for them. The presented work is seen as a primary concept and proof of principle, and can be expanded to include a larger patient population in subsequent work with a higher confidence. The preliminary kind of the work was mentioned in the Discussion and Conclusions.

Reviewer 3 Report

The major limitation of the current study is the small sample size. 

Author Response

The small sample size is related to the limitations in the selection of the required number of clinical patients who would have a confirmed diagnosis but did not have treatment at the time of sampling and who could be tracked after starting therapy over a long time. Under these conditions, it is not possible to redo the experiment with a larger sample set in a short time.

Round 2

Reviewer 2 Report

My first remark was not clear, I am sorry, I would like to see a graph for the patients used, with on the X-axe the time to treatment, for the 3 proteins. I still think even a small group of patients for validation would strengthen the paper considerably, and would make my second remark obsolete, but I understand it is not possible.